# Human resources challenges in the management of diabetes and hypertension in Mozambique

**Tavares Madede** [1,2]*, **Elzier Mavume Mangunyane**[1], **Khátia Munguambe**[1], **Valério Govo**[3], **David Beran**[4], **Naomi Levitt**[2], **Albertino Damasceno**[3,5]

**1** Department of Community Health, Faculty of Medicine, University Eduardo Mondlane, Maputo, Mozambique, **2** Chronic Disease Initiative for Africa, Department of Medicine, Faculty of Health Sciences, University of Cape Town, Cape Town, South Africa, **3** Research Unit, Department of Medicine, Maputo Central Hospital, Maputo, Mozambique, **4** Division of Tropical and Humanitarian Medicine, University of Geneva and Geneva University Hospitals, Geneva, Switzerland, **5** Faculty of Medicine, Eduardo Mondlane University, Maputo, Mozambique

* tavares.madede@uem.mz

**Data Availability Statement:** All relevant data are within the manuscript.

**Funding:** This study forms part of a thesis for a Doctoral degree funded by the COHESION Project

## Abstract

### Background

The major burden of non-communicable diseases (NCDs) globally occurs in low-and-middle-income countries, where this trend is expected to increase dramatically over the coming years. The resultant change in demand for health care will imply significant adaptation in how NCD services are provided. This study aimed to explore self-reported training and competencies of healthcare providers, and the barriers they face in NCD services provision.

### Methods

A qualitative design was used to conduct this study. Data was collected through semi-structured interviews with government officials within the Mozambican Ministry of Health, district health authorities, health facility managers, and health providers at urban and rural health facilities of Maputo, in Mozambique. The data was then analyzed under three domains: provider´s capacity building, health system structuring, and policy.

### Results

A total of 24 interviews of the 26 planed with managers and healthcare providers at national, district, and health facility levels were completed. The domains analyzed enabled the identification and description of three themes. First, the majority of health training courses in Mozambique are oriented towards infectious diseases. Therefore, healthcare workers perceive that they need to consolidate and broaden their NCD-related knowledge or else have access to NCD-related in-service training to improve their capacity to manage patients with NCDs. Second, poor availability of diagnostic equipment, tools, supplies, and related medicines were identified as barriers to appropriate NCD care and management. Finally, insufficient NCD financing reflects the low level of prioritization felt by managers and healthcare providers.

financed by the Swiss Agency for Development and Cooperation (SDC) (www.swissaid.ch) and the Swiss National Science Foundation (SNF) (www.snf.ch), under the funding scheme r4d - Swiss Programme for Research on Global Issues for Development, grant number #160366. The funders had no role in study design, data collection and analysis, decision to publish, or preparation of the manuscript.

**Competing interests:** The authors have declared that no competing interests exist.

## Conclusion

There is a gap in human, financial, and material resources to respond to the country's health needs, which is more significant for NCDs as they currently compete against major infectious disease programming, which is better funded by external partners. Healthcare workers at the primary health care level of Mozambique's health system are inadequately skilled to provide NCD care and they lack the diagnostic equipment and tools to adequately provide such care. Any increase in global and national responses to the NCD challenge must include investments in human resources and appropriate equipment.

## Introduction

Non-communicable diseases (NCDs) are considered a global health challenge. The World Health Statistics (2023) estimated that of the 55.4 million deaths from all causes in 2019, 74% deaths were from NCDs [1]. Global deaths projections indicate that by 2048 there will be nearly 90 million annual deaths, with a 90% increase in the deaths attributable to NCDs from 2019 [1].

The major burden of NCDs occurs in low-and middle-income countries (LMIC) [2]; which registered 77% of the approximately 41 million deaths attributed to NCDs in 2021 [3]. LMICs, including countries in Sub-Saharan Africa (SSA), are expected to register a greater increase of the NCD burden due to their population ageing and rapid urbanization [4]. This is a major problem as LMIC healthcare services are predominantly oriented towards acute conditions and infectious diseases. Globally, international donors have the notion that investment should first go to communicable diseases before they start focusing on NCDs, and locally, health systems base their prevention and control measures on infectious diseases [5].

The demands imposed by the rising NCD burden on LMIC health systems including diagnosis, management and treatment, will require a reorientation of healthcare systems to improve their responsiveness and address the new challenges accordingly [6]. Overall health system strengthening has been identified as a critical strategy in addressing the NCD burden [6, 7]. Health system strengthening requires allocation of adequate numbers of qualified human resources (HR), one of its key building blocks. One of the most pressing challenges of LMICs' health systems is the HR scarcity. In 2019 it was estimated that in SSA there were 2.9 physicians per 10,000 inhabitants in contrast to 6.5 physicians per 10,000 inhabitants in South Asia. Similarly, nurse and midwifery personnel ranged from 9.7 per 10,000 inhabitants in South Asia to 18.3 per 10,000 inhabitants in SSA [8]. These figures are up to 5 to 10 times less than the estimates for high-income countries' physicians (33.4 per 10,000 inhabitants) and 6 to 11 times fewer nurses and midwifery personnel (114.9 per 10,000 inhabitants) for the same year [8]. This scarcity is further aggravated by factors such as training, mentoring and other supporting measures from health systems, which are largely shaped by health sector priorities [9].

In Mozambique, NCD have received some prominence since the early 2000s when reports started to identify a significant burden that has worsened over time. For example, in 2001, mortality from NCD ranged from 13 to 24% [10], then rose to 28% in 2012 [11]. Similarly, the estimated prevalence of hypertension and type 2 diabetes increased from 33% to 39% and 2.9% to 7.4%, respectively, from 2005 to 2015 [12–15]. This increase in prevalence was accompanied by a deterioration in other NCD health parameters, such as fewer people being 1) aware of their disease, 2) being in treatment, and 3) with their diseases controlled [14–16].

In recognition of the burden of NCD in Mozambique, particularly of hypertension and diabetes, NCD were included in national level health and priority policy documents, such as the Poverty Reduction Action Plan (PARPA II), which declared NCD as a priority in 2006, and later in 2008, the first National Strategy for Prevention and Control of NCD [17] was developed. The strategy aimed to reduce exposure to NCDs risk factors and associated morbidity and mortality by increasing awareness about NCDs, strengthening and integrating NCD-related training, improving access and quality of NCD prevention and care services, and strengthening NCD surveillance, monitoring and evaluation [10]. However, the reported HR shortage in Mozambique's health system is severe, with less than one doctor per 10,000 people and approximately three nurses per 10,000 people [18, 19], which poses a significant challenge to the effort to strengthen the health system.

This study's aims were: (i) to explore self-reported training and competencies of NCD-related healthcare providers, and (ii) to assess the perceived challenges posed by the current health system structuring in the health sector context of the country. By documenting the perceived challenges to managing diabetes and hypertension, the study team expects to contribute to informing the healthcare workers' training requirements and healthcare services structuring to provide NCD-related services tailored to the needs of Mozambique's and similar health contexts.

## Material and methods

### Study design

We conducted a qualitative study between February and March 2017. This study was comprised of semi-structured interviews with government officials within the Mozambican Ministry of Health, district health authorities, health facility managers, and health providers. The interviews focused on exploring factors related to the training healthcare workers received, the competencies acquired, and the existing resources to support NCDs (diabetes and hypertension) prevention and management at the primary health care (PHC) level. An interview guide was developed based on the literature and integrated the following topics: healthcare workers, diagnostic tools and technologies, medicines and guidelines, referral system and other type of support, and NCD prioritization. The consolidated criteria for reporting qualitative research (COREQ) [20] was used for this study reporting.

### Study setting

This study was conducted at the national level and in two districts in Southern Mozambique. The districts were Nlhamankulu and Moamba, with suburban and rural characteristics. In both districts, two primary health facilities were selected from the 15 existing facilities. In Nlhamankulu District the facilities selected include Xipamanine and Chamanculo health centers; and in Moamba District the facitiies selected include Moamba and Sábiè health centers. Healthfacility selection was based on their characteristics, such as location (suburban vs. rural areas), easy access, catchment area, target population, and the number of patients assisted with NCD. Primary health facilities constitute the entry point to the national health services, offering mainly promotive, preventive, and curative services for minor diseases, such as acute respiratory illness and diarrhea without dehydration. In addition, health programs for malaria, HIV/AIDS, sexually transmiited infections (STI), and tuberculosis are primarily implemented at this level. These primary health facilities comprise 95% of the national health facility network and are estimated to cover 80% of the population with health services [18]. However, despite this high proportion, they only have 36% of the total health care providers [21].

Nlhamankulu District has seven health facilities, comprised of two general hospitals and five health centers. These seven facilities have 209 health care providers, including mid-level technicians, general practitioners, and specialist doctors. Of the 209 providers, 80% are distributed between the two selected study facilities, which combined provide health services to 62% of the districts 130,000 inhabitants. Per annual district health reports, the main reported health problems in 2018 were malaria, diarrhea, malnutrition, HIV/AIDS and tuberculosis [22]. The report did not include data on hypertension and diabetes.

Moamba District has ten health facilities, comprised of nine health centers and one health post, all primary level, with 102 health care providers and 25 community health workers. Of the 102 providers, 66% are distributed between the two study facilities, covering almost 50% of the district´s 90,000 inhabitants [23]. As in Nlhamankulu, the main health problems reported in 2018 were malaria, diarrhea, malnutrition, HIV/AIDS and tuberculosis. As of 2018, there were 963 cases of hypertension and eight cases of diabetes reported in Moamba [23].

In general, at the PHC level, NCD related services are provided by three types od cadres. Doctors [24] and health technicians [25], who have more clinical focus, including diagnosis and treatment, and nurses [26], who are more focused on general care to patients, not necessarily directed to a specific type of disease. The pre-service training for all categories have a number of hours dedicated to practice at the health facilities, where students apply the theoretical knowledge, which is more than two times the hours dedicated to learning theory at the classroom [24–26]. Practical learning components within health facilities are shaped by the health sector priorities [9]. Although both infectious diseases and NCDs co-exist, attention is much more directed towards responding to the former, more specifically to HIV, tuberculosis and malaria, as the most prioritized [5, 6, 10].

## Data collection procedures and sampling

**Semi-structured interviews.** Semi-structured interviews were conducted by five researchers (two females and three males), each with a minimum of a bachelor's level education in anthropology, biology, or medicine, and all of them had previous experience conducting qualitative interviews ranging from 2 to 15 years. Interviews were over two months, between February and March 2017.

The researchers had a three-day refresher training on qualitative methods and discussed the study protocol to gain familiarity. Interviews were audio recorded, and the interviewers took notes during the interview. Interviews were conducted in Portuguese because all participants could communicate fluently in Portuguese. Interviews lasted between 30 and 60 minutes and continued until all topics in the interview guide were covered and no new issue emerged (theoretical saturation). The interviews were anonymized.

In order to attain a complete understanding of the breadth and depth of factors that may influence NCDs-related health professionals' qualifications and their capacity to provide optimal care, a variety of stakeholders were interviewed. A total of 26 MOH health workers and managers at various levels were invited for interviews, and two were unavailable due to concurrent agenda, resulting in 24 interviewees conducted and a 92% response rate. Three national level, four district level, and seventeen health facilities levels managers and providers were interviewed. We used a convenience approach to select respondents, considering a minimum of 6 months at the current position or role, deemed enough to understand the context within which they operate. At the national level, interviewed individuals were viewed as key opinion leaders from the Ministry of Health who could understand the contextual factors leading to the development of the policy documents and the development of the health workforce. At the district level, health authorities (health directors and chief medical officers) and focal

points for the different NCD programs were interviewed to explore their experiences regarding the availability of staff and the type of conditions in place to provide specific health care services at that level. Lastly, at the health facility level, managers, wards-in-charge or sectors-in-charge personnel and health practitioners were interviewed to explore their daily experiences in implementing NCD-related activities.

## Data management and analysis

The audio recordings were transcribed verbatim in Portuguese and NVivo software (*NVivo 12*, *QSR International)* was used to facilitate coding (thematic analysis).

A codebook with a predetermined list of themes (categories) and sub-themes (sub-categories) was developed and a deductive perspective used for data analysis. These themes were identified by drawing on the adapted model shown below (Fig 1), which puts healthcare providers' capacity building as the core asset for adequate provision of NCD-related care. We used a phenomenological approach to explore healthcare providers and their manager's perceptions of how the structure of the health system influences how they are prepared to respond to the NCD service's needs, and the additional investment on diagnostic tools, equipment, supplies, and medicines necessary to provide quality services under the policy prioritization context it functions.

The lead and second authors conducted a parallel coding guided by the developed codebook. After concluding the process individually, they jointly compared the coded transcripts. They reviewed all coded transcripts to reach a consensus on which quotes to present that best captured respondents' perspectives and responded to the analysis domains. After being sorted by theme (category), data was summarized, synthesized, and abstracted in each category. Translation of the relevant sections of the transcriptions to English language was done at the time of writing this manuscript.

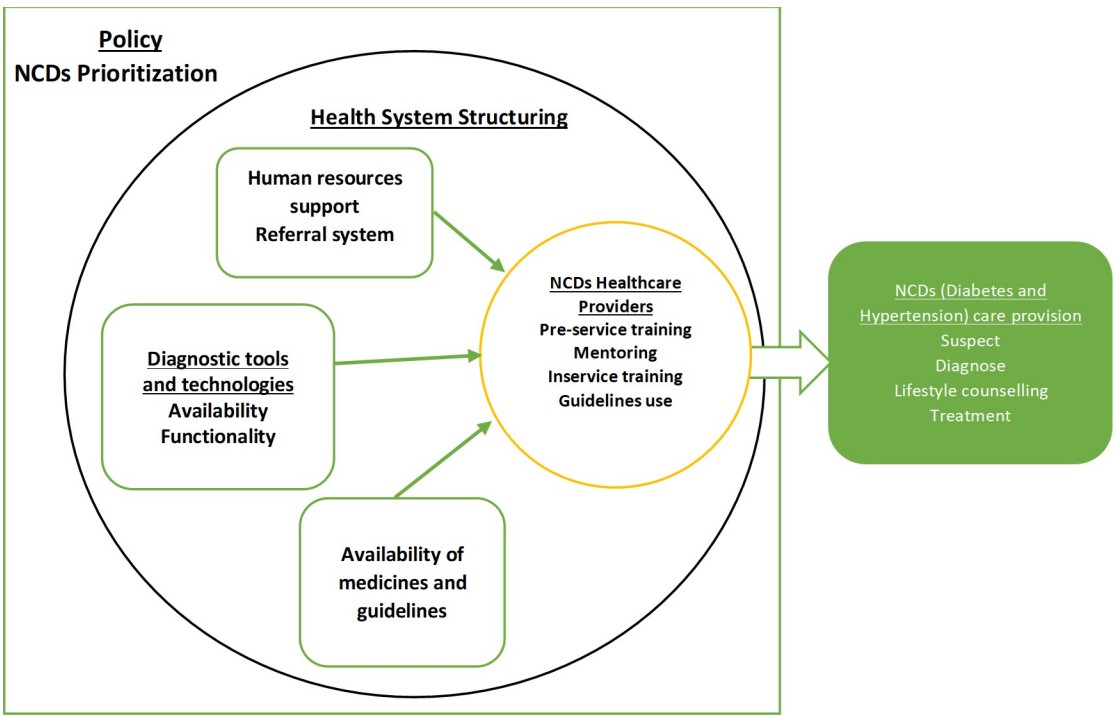

**Fig 1. Study's data analysis model.**

### Ethics approval and consent to participate

This research was conducted in accordance with the Declaration of Helsinki for research involving human participants. The Institutional Bioethics Committee of Health of the Faculty of Medicine and Maputo Central Hospital in Mozambique approved the research project with the reference: CIBS FM&HCM/55/2016. Written informed consent was obtained from each participant after being taught about the voluntary nature of the study and that they could withdraw at any time without consequences. Interviewers introduced themselves and then provided the interviewees with an information sheet about the study and gave each participant an informed consent document to read and sign. Time was provided for asking and answering questions about the study before the consent form was signed. They were also informed about the risks and benefits of participating in the study.

## Results

For both diseases, hypertension, and type 2 diabetes, we describe the findings following the three domains identified on the model: providers capacity building, health system, and policy. Within each domain we identified common barriers (**Table 1**) across the two districts and all four health facilities, which were: (i) inadequacy of NCD's healthcare providers training and poor guidance to enhance providers skills; (ii) disrupted referral system, poor availability of diagnostic meansand medicines; and (iii) low NCDs' priority.

### Healthcare providers capacity building

**Inadequacy of NCD's healthcare providers training.** *Pre-service training inadequate for NCD-related needs*. The NCDs related competencies acquired during pre-service training are limited as evidenced by the documents review presented above and supported by the interviews. For example, six of the eight healthcare workers interviewed referred to the need to consolidate and broaden their NCD-related knowledge, to improve their capacity to treat their patients and be able to teach them how to use non-therapeutic measures confidently.

> *P1: ". . .one of the main problems we have is that clinicians can only provide limited information to patients. . .they are able to tell that they (patients) have the disease and prescribe the*

**Table 1. Structured themes and subthemes from the interviews by level of analysis.**

| Analysis Level | Themes | Sub-themes |
|---|---|---|
| Healthcare providers capacity building | Inadequacy of NCD's healthcare providers training | Pre-service training inadequate for NCD-related needs |
| | | Lack of in-service training opportunities |
| | Poor guidance to enhance providers skills | Lack of NCDs' related guidelines and algorithms |
| Health System Structuring | Disrupted referral system | Incomplete patient follow-up through the referral system |
| | Poor availability of diagnostic means and medicines | Shortage of functioning diagnostic tools and equipment |
| | | Lack of local laboratories' capacity |
| | | Inconsistent availability of laboratory supplies |
| | | Limited quantities of available medicines |
| Policy | Low NCDs' priority | Resources planning incompatible with the needs |
| | | Poor investment on NCDs related services |

*first line medicines but are unable to explain about appropriate diet and lifestyle and long-term measures or even why patients have to live on drugs for their entire life.*

Likewise, health managers and non-physician providers recognized the providers' deficiency to diagnose and adequately treat NCDs and expressed the need for acquiring additional skills due to limited pre-service training exposure to NCDs' core subjects. The lack of skills delays the patients' diagnosis and affects the appropriate NCDs management, mainly diabetes, at the PHC.

*P2: ". . .the training (pre-service) and information I have on diabetes and hypertension is not enough to take care of patients. . . I feel like I need more."*

*FP1: ". . .hypertension is not a problem amongst us here. . .providers treat and follow-up patients with no complaints. . .but we have many difficulties with diabetes. . ."*

*M1: "I notice that they (providers) have a basic knowledge, but not so thoughtful that it can ensure that they suspect or confirm all existing cases (of diabetes) here."*

This has greatly contributed to the observed NCDs related healthcare providers shortage described in official documents from and surveys conducted by the MOH. This shortage was evident during the study's data collection, observing that the four health facilities visited had one or two non-physician clinicians (providers who complete 2 to 3 years of clinical training after secondary education) [27], who partially dedicated their time to NCDs in addition to the doctors, who were available in only two health facilities but not always on duty due to others non-clinical responsibilities.

*Lack of in-service training opportunities.* Respondents noted that short in-service training is required to refresh healthcare providers' skills to manage NCDs. All eight providers interviewed recognized the benefits of having short in-service refresher courses to improve their skills in managing NCDs,

*P4. "In my opinion, having in-service training would help us greatly improve our approaches towards patients with diabetes and hypertension. . ."*

and, to ensure updated approaches and the use of new available techniques and diagnostic means and tools.

*P6: ". . .the only time I was trained for these two diseases was during my pre-service training. . .I think I need more. . .there are new approaches, and we are back on time. . ."*

However, they reported long periods without having the opportunities to receive such training, as opposed to the periodic sessions witnessed within the HIV, TB, and Malaria programs.

*P5: ". . .since I have started working at this Centre (more than 5 years ago) we never received any type of in-service training for hypertension or diabetes. . ."*

The reported gaps in pre- and in-service training collectively contribute to low confidence of healthcare workers in managing NCDs. Respondents from two health facilities highlighted the delay in diagnosing patients resulting from their fear of the complexity of managing diabetes.

P3: "...We would always avoid doing screening for diabetes, patients would present signs and symptoms, but we would ignore them and look for other diagnoses... when we decided to measure the first glycaemia it was around twenty-five, maybe the patients were already developing the disease and we ignored it until that it ended up reaching that extreme..."

**Poor guidance to enhance providers skills.** The managers interviewed at all levels suggested a degree of negligence from frontline providers by not using the diabetes and hypertension management guidelines developed to help them improve their capacity to diagnose and treat such conditions.

M4: "...we have developed specific guidelines (for diabetes and hypertension) and sent them out to all provinces, however every time we do supervision visits, we find them stored in drawers under their (providers) desks..."

On the other hand, the healthcare workers from the rural district were unanimous about not receiving any clinical guideline and, therefore, stated this resulted in either not having the confidence of managing the cases appropriately or contributing to inconsistent procedures between clinicians to manage the same condition.

P2: "...I think one aspect that should be improved are the procedures...they are not uniform and therefore each clinician proceeds his/her own way...we need to avoid different approaches for the same problem ..."

## Health system organization

**Disrupted referral system.** *Incomplete patient follow-up through the referral system.* Health workers and managers at the PHC level pointed out that once patients are referred to higher levels for acute management, it becomes more challenging to follow them up because they do not receive appropriate support to monitor patients after being assisted by a specialist. Additionally, they lack resources and medicines since this level of care does not have the same resources as is observed with higher levels, which interrupts the patient follow-up and jeopardizes the trust of the service.

P8: "...our providers have difficulties to adjust medications at this level, so we send them to higher levels, however they get lost when they come back and need different medicines than those that we have at this level..."

**Poor availability of diagnostic means and medicines.** *Shortage of functioning diagnostic tools and equipment.* National-level managers indicated that all health facilities in the country consistently received equipment for screening and testing,
however, health facilities' managers and healthcare workers reported an equipment shortage and malfunctioning.

P3: "...we don't have enough working equipment...it should be normal for us to have hypertension devices for each technician or per cabinet and a stethoscope, but we don't...we only work with two devices for this entire health facility..."

M1: "...regarding glycaemia, we have a chronic problem, when we have strips, they are not suitable to the glucometers we do have, and vice-versa..."

*P1: "...when we talk about hypertension, the major problem we have here is related to the equipment (sphygmomanometers)...we have the digital type, which uses batteries, and often times we don't have batteries..."*

*Lack of local laboratories' capacity.* According to healthcare workers, there is lack of laboratory capacity, which demoralizes both providers and the patients to seek diabetes-related care due to long waiting time, limited availability of supplementary exams, and the need to expend additional money to return to the health facility only to collect the result.

*P6: "...it is difficult to follow-up diabetic patients when we only use glycemia...other means such as glycated hemoglobin, which is ideal for these cases, is never available..."*

*P4: "...we don't process glycemia here, so we have to send the blood sample to Matola Provincial Hospital (referral hospital) ...sometimes we have the results the following day, but others not...so we ask the patients to return after two or three days. However, some of them do not return, saying they do not have money for transport...they actually get upset when they come and don't find their results......"*

*Inconsistent availability of laboratory supplies.* A healthcare worker from one of the health facilities highlighted the generalized laboratory supplies stock-outs often with a long waiting time to replenishment.

*P7: "...we have had difficulties, sometimes in the lab we have reagents stockouts...it's normal for us to stay a week without blood glucose reagents, so we send the patients to Chamanculo to do it there, although sometimes they also have the same problems..."*

*Limited quantities of available medicines.* All managers at different levels pointed out that the system was experiencing varied levels of essential NCDs medicines stockouts. District-level managers associated the medicines shortage with the absence of health partners interested in NCDs as opposed to what they observe for communicable diseases such as HIV and TB, which have enough resources and fewer reported stockouts.

*M3: "...most of the times we have to work with almost no resources, not even basic (first line) medicines and that makes it difficult for us to diagnose and treat these diseases...if we had the same resources we have for HIV and TB we would do better..."*

## Policy

**Low NCDs' priority.** *Resources planning incompatible with the needs.* Respondents raised this issue, mentioning that low prioritization of NCDs has negatively influenced the quality of their response, resulting in poor resource availability due to insufficient financing. For example, due to limitations in BP cuffs and blood glucose strips, participants mentioned that they only measure parameters like blood pressure and blood glucose on selected patients, such as those 35 years and older, based on their perceived higher risk.

*PF2: "...with the equipment problem we have we try to make a rational use of the existent, for example for hypertension we measure blood pressure of patients (health facility users) who are old and only rarely we measure those under the age of 35 if they have any complaints that relate to the disease...for diabetes we have many more difficulties..."*

The reported scenario is opposed to other health programs, such as those related to HIV and TB, which have enough resources and rarely register stockouts.

*P8". . .most of the times we have to work with almost no resources and that makes it difficult for us to diagnose and treat these diseases. . .if we had the same resources we have for HIV and TB we would do better. . ."*

Furthermore, while it is rare to notice stockouts of HIV and TB related medicines, the same does not hold for diabetes and hypertension. These patients, who should collect their medicines every month, in the best case scenario, collect them for half that period, when they are available, to allow more people to access medication. Consequently, the cost of collecting medicines for these patients is exacerbated by the combined costs of transport and opportunity, given the uncertainty of its availability. Some patients opt to by in private pharmacies through their own means or loans.

*FP2: ". . .it is normal to have limited stocks of anti-hypertensives in this health facility. . .we have to manage what we have, so instead of giving enough for the whole month, we give small portions and ask the patients to come back to see if we received more medicines later. . ."*

*P8: ". . .patients with conditions buy at the private pharmacy, but we only have one pharmacy here in the countryside, and sometimes they also run out of stock. . ."*

*Poor investment on NCDs related services.* Some healthcare workers understand the observed level of investment in HIV and related services as a clear message about where their focus should be, disregarding the importance of other health conditions including NCDs.

*P5: ". . .I think we have cases of diabetes and hypertension in the district, however we don't make so many diagnosis because our main focus is on HIV patients. . ."*

## Discussion

Although the reviewed cadres' training curriculum incorporates NCD-related content, the findings from this study suggest that nurses and non-physician clinicians rarely engage in diagnosing and managing NCDs at the PHC level due to limitations in their pre-service training. The healthcare sector of Mozambique lacks diagnostic equipment, tools and supplies, and medicines which constitutes an additional barrier to healthcare providers ability to manage NCDs appropriately; and, there is a lack of guidelines and training in their use to improve providers' skills.

The training models adopted by health training institutions in LMIC, which have a significant practical component completed within the health facilities, are shaped by the health sector´s priorities [9]. Although both infectious diseases and NCDs co-exist, attention is much more directed towards responding to the former, more specifically to HIV, tuberculosis and malaria. This is frequently driven by donor funding, as shown by the high level of investment and the amount of existing supporting resources (guidelines, diagnostic equipment and supplies, medicines, patient registries, to name some) in health facilities [28–30]. Therefore, health students are much more likely to practice and learn about infectious diseases, while the NCDs lag behind. Secondly, there is a well documented investment in short in-service trainings targeted towards doctors, non-clinician health technicians, and nurses that mainly focuses on three priority diseases, HIV, tuberculosis and malaria [29]. Funders are interested in this type

of training to show results within the time frame of specific grants by continuously adapting to the local needs, ensuring providers are up-to-date on the latest approaches and international guidelines and improving their performance against pre-determined indicators [9, 29]. In Mozambique, external institutions supporting the large programs for HIV, TB and malaria will frequently provide professionals with technical skills that are available to provide mentorship and guidance to the frontline providers in case they need it [9]. However, this same level of technical support has not yet prioritized NCD care.

Apart from the healthcare workers' limited exposure to NCD-related knowledge and practice during their pre-service training, after their integration into the healthcare system they face additional challenges. The necessary equipment and tools to support them with their activities were neither sufficient nor adequate to help them improve the quality of services they offer. For example, the diagnostic equipment required to screen for and diagnose both diabetes and hypertension was revealed to be unreliably available [31]. When available, the equipment often did not correspond to the actual needs of patients, such as those needing extra-large BP cuffs even though obesity is a common risk for diabetes [15].

Moreover, providers could benefit from using MOH produced clinical guidelines to improve their management of diabetes and hypertension. Apart from supporting healthcare providers, the guidelines serve other purposes, including harmonization of clinicians' approaches [32], improving the confidence of clinicians who are unsure on how to proceed, and reversing clinicians' beliefs adapted to outdated practices [33]. However, some mid-level providers perceive their use as an additional burden to the providers' complex working environment.

Lastly, the NCDs referral system does not consider the uneven distribution of resources between levels of care, further aggravating the perceived barriers at the PHC level. The nature of the NCDs discussed here requires the same core type of resources for their management when there are no complications. Therefore, it would be needed to ensure the availability of similar types of resources, including qualified providers, laboratory equipment and consumables, and medicines, at the different levels, including the PHC where the demand is higher and there are competing priorities.

Although there are apparent efforts from the government to change the current scenario, early detection, treatment, and control of diabetes and hypertension is unsatisfactory in Mozambique's health care system, similar to other countries in the African region, which results in frequent and severe complications and sequelae, along with premature deaths [14, 16, 34]. For more than a decade now, Mozambique has taken decisive steps in recognizing the importance of NCDs in the national epidemiological profile by highlighting them in the government's guiding documents, developing NCDs specific strategic plans, and investing in medicines and laboratory consumables for NCDs to provide them to the public at no cost. The introduction of reforms in the system to tackle the NCD burden and ensuring a thorough implementation of the plans would enable an appropriate response from the health services, but this is not visible. A similar observation was made in a recent study by Heller et al. [35], who concluded that policies, strategies and effective interventions to NCD control exist, but their applicability is questionable, especially in LMICs. In fact, the National Health Service has theoretically adopted integrated management of diseases to allow a comprehensive provision of health care. However, most of the programs, including HIV, TB, and malaria, still function vertically, which further affects the availability of other less prioritized services [36], such as the NCDs. This leads to uneven access to resources and services, contributing to the system's inefficiencies [36, 37].

There were some limitations in this study. First, the NCD-related skills reported relied mainly on self-reported assessment, which could affect the reliability of the study findings. A

strength was that we used data triangulation to enhance the study findings, validity, and credibility. Second, in Mozambique there are several independent health training institutions graduating students, that are then placed within the health system which may mean that some health professionals were not trained under any of the assessed curricula. Further, we did not conduct an in-depth evaluation of the content of the training curricula since this was not the aim of the study. We only visited a limited number of health facilities for this study, all located in a relatively resources favored provinces (Maputo Province and City). This may have biased our findings by underestimating the level of resource scarcity in the PHC-level health facilities.

The health system needs to adapt to offer effective treatment, self-management support, and regular follow-up to NCD patients [38–40]. Some tasks, such as counselling for self-management, are needed to manage NCDs but are only learned by practice. Unfortunately, most of the time, the workload of clinicians does not allow them sufficient time to provide adequate counselling to patients and their families. Tasks like this can be shifted to non-clinicians, such as lay counsellors, who have been successfully used in other health programs. Thus, comprehensive NCD care can be provided to patients by providing appropriate training and evidence-based skills, as well as organizing health care providers into team care models for managing chronic health problems [38]. This approach would reduce the number of trained professionals needed to achieve satisfactory results of NCDs control [41–43] by restructuring the few existing providers and ensuring an even resources distribution among the types of health problems facing the healthcare system.

## Conclusions

Mozambique´s health strategic and policy documents reflect the concern at national governmental level about NCDs, however this has not translated into practice. There is a gap in health resources, including human, financial, and material, to respond to the needs faced by the country's health system. This is more significant for NCDs as they need to compete with the major infectious diseases, which overall are better funded by external partners. The healthcare workers available to provide NCD-related care and management at the PHC level of Mozambique's health system are inappropriately skilled. In addition, these professionals face a lack of diagnostic equipment and tools to adequately respond to NCD-related needs, particularly at the primary care level. Any increase in global and national responses to the NCD challenge must include investments in human resources and appropriate equipment.

## Acknowledgments

The study team would like to thank the following individuals and institutions: the Mozambique MOH; the Maputo Province and City, and Nlhamankulu and Moamba Districts Directorates of Health; the health facilities managers and providers for granting permission to conduct the study and for their personnel contributions through interviews; the COHESION study group for working on different stages of the study conceptualization, field work preparation, data collection and data management; Dr Troy Moon for editing and polishing the manuscript and for providing insightful comments to improve its clarity and readability.

## Author Contributions

**Conceptualization:** Tavares Madede, Khátia Munguambe, David Beran, Albertino Damasceno.

**Formal analysis:** Tavares Madede, Elzier Mavume Mangunyane.

**Funding acquisition:** David Beran.

**Investigation:** Tavares Madede, Khátia Munguambe, Valério Govo.

**Methodology:** Tavares Madede, Elzier Mavume Mangunyane, Valério Govo.

**Supervision:** David Beran, Naomi Levitt, Albertino Damasceno.

**Validation:** Naomi Levitt.

**Writing – original draft:** Tavares Madede.

**Writing – review & editing:** Tavares Madede, Elzier Mavume Mangunyane, Khátia Munguambe, Valério Govo, David Beran, Naomi Levitt, Albertino Damasceno.

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
