## [Decision Letter · Decision Letter 0]

21 Nov 2023

PONE-D-23-22818Human resources challenges in the management of diabetes and hypertension in MozambiquePLOS ONE

Dear Dr. Madede,

Thank you for submitting your manuscript to PLOS ONE. After careful consideration, we feel that it has merit but does not fully meet PLOS ONE’s publication criteria as it currently stands. Therefore, we invite you to submit a revised version of the manuscript that addresses the points raised during the review process.

We look forward to receiving your revised manuscript.

Kind regards,

Joel Msafiri Francis, MD, MS, PhD

Academic Editor

PLOS ONE

Journal Requirements:

Reviewers' comments:

Reviewer's Responses to Questions

**Comments to the Author**

1. Is the manuscript technically sound, and do the data support the conclusions?

Reviewer #1: Yes

Reviewer #2: Yes

2. Has the statistical analysis been performed appropriately and rigorously? 

Reviewer #1: N/A

Reviewer #2: Yes

3. Have the authors made all data underlying the findings in their manuscript fully available?

Reviewer #1: Yes

Reviewer #2: Yes

4. Is the manuscript presented in an intelligible fashion and written in standard English?

Reviewer #1: Yes

Reviewer #2: Yes

5. Review Comments to the Author

Reviewer #1: Overall

An interesting paper dealing with a relevant health issue. One may challenge the design of the study that incorporated components that were not purely qualitative. The theoretical framework was not clearly stated nor the data management. The results were not adequately presented.

Introduction

In the first paragraph, one might challenge that the references used are outdated. Could the authors use newer publications on the issues raised?

Methods

Study design

One might challenge the place of document review and the observation that render the study not purely qualitative. I would suggest to use the information gathered from these two approaches only for the description of the study setting.

Theoretical framework

The methodological orientation was not clearly stated.

Study population

The justification of the sample size was not stated. Nor the number of the persons invited for the study. What was the response rate?

Data collection

The experience of the five researchers in qualitative research is not expressed.

The duration of the interview was not stated. What guided the authors to close the study? Data saturation?

Data analysis

The role of the researchers in the data coding was not expressed. It is not clear how many persons did the data coding. Did the researchers participate in data coding? How did the authors acted to ensure the trustworthiness of their findings? Did they did member checking or peer review?

What kind of thematic analysis did the authors use?

Results

Identification of the themes

It seems that the pre-identified themes are the same with that emerged from the interviews. Was there none deviation from what was described before the data collection?

In the table 2, some themes had only one sub-themes, could the authors reconsider the grouping of the themes?

Presentation of themes

If one may encourage thick description, I would discourage to include in-text citations of references in this part.

References

For those in other languages than English, put their titles in English in square brackets.

Reviewer #2: The manuscript is very well written with a clear abstract, introduction, methodology, results and discussion. The study objects have been answered diligently and conclusions match the results.

The grammar is excellent and easy to follow. Authors should correct one spelling error in line 440, replacing "by" to "buy

6. PLOS authors have the option to publish the peer review history of their article (what does this mean?). If published, this will include your full peer review and any attached files.

Reviewer #1: **Yes: **Jean-Pierre Fina Lubaki

Reviewer #2: **Yes: **Godfrey Mutashambara Rwegerera

---

## [Author Response · Author response to Decision Letter 0]

8 Jan 2024

Thank you very much for your valuable comments ans suggestions. All relevant answers are embodied in the response do reviewers which addressed point-by-point all comments and suggestions.

---

## [Editor Report · Decision Letter 1]

11 Jan 2024

Human resources challenges in the management of diabetes and hypertension in Mozambique

PONE-D-23-22818R1

Dear Dr. Madede,

We’re pleased to inform you that your manuscript has been judged scientifically suitable for publication and will be formally accepted for publication once it meets all outstanding technical requirements.

Kind regards,

Joel Msafiri Francis, MD, MS, PhD

Academic Editor

PLOS ONE
---

## [Editor Report · Acceptance letter]

20 Mar 2024

PONE-D-23-22818R1 

PLOS ONE

Dear Dr. Madede, 

I'm pleased to inform you that your manuscript has been deemed suitable for publication in PLOS ONE. Congratulations! Your manuscript is now being handed over to our production team.

Kind regards, 

on behalf of

Prof Joel Msafiri Francis 

Academic Editor

PLOS ONE